# SPACE FOR COPD delivered as a maintenance programme on pulmonary rehabilitation discharge: protocol of a randomised controlled trial evaluating the long-term effects on exercise tolerance and mental well-being

Khaled A Alqahtani [1,2] Charlotte Gerlis,[3] Claire M Nolan [4] Nikki Gardiner,[3] Ala Szczepura [5] William Man,[4,6] Sally J Singh,[2,3] Linzy Houchen-Wolloff [2,3]

For numbered affiliations see end of article.

**Correspondence to**
Dr Linzy Houchen-Wolloff;
Linzy.Houchen@uhl-tr.nhs.uk

## ABSTRACT

**Introduction** The benefits achieved during pulmonary rehabilitation (PR) are known to be sustained for 6–12 months after the initial programme. Several maintenance trials have been conducted but were heterogeneous in terms of duration, frequency and labour cost. There is no consensus on one best strategy. SPACE FOR COPD (Self-management Programme of Activity, Coping and Education for Chronic Obstructive Pulmonary Disease) is a home-based self-management programme, which has been shown previously to be effective in primary and secondary care settings and is to be tested here as a maintenance programme. The aim is to evaluate the efficacy of the SPACE FOR COPD programme (manual and group sessions), on exercise tolerance and mental well-being, compared with usual care following PR in patients with COPD.

**Methods and analysis** A prospective, multicentre, single-blinded randomised controlled trial requiring 116 participants with a clinical diagnosis of COPD who have finished PR within 4 weeks will be randomised 1:1 to either a usual care group or a SPACE FOR COPD programme group. The intervention comprises a home-based manual and 4, 2-hour group sessions adopting motivational interviewing techniques over 12 months. The primary outcome is endurance capacity measured by the Endurance Shuttle Walking Test at 12 months. Secondary outcomes are: maximal exercise capacity, health-related quality of life, mood, patient activation, physical activity, lung function and healthcare costs. The measures will be taken at baseline, 6 and 12 months. Patient interviews and staff focus groups will be conducted to explore barriers, facilitators and views about the intervention at the end of the study. A framework analysis will be used for the interpretation of qualitative data.

**Ethics and dissemination** The trial was granted ethical approval from Health Research Authority and Health and Care Research Wales (HCRW19/EM/0267 on 10 October 2019). Results will be made available to all stakeholders through a dissemination event, conferences and peer-reviewed publications.

---

### Strengths and limitations of this study

► This is a novel maintenance trial that uses an evidence-based home-based manual integrated with motivational interviewing skills to promote long-term self-management and behavioural change.

► Patients with chronic obstructive pulmonary disease who are registered in our patient and public involvement group were invited to review the protocol and two were involved in every aspect of developing the project.

► This is a multicentre trial, therefore the results may be more generalisable than a single centre.

► The programme is limited to English speakers only who can read at the level of an 8 year old. This may, therefore, exclude those who do not read in English or have low literacy.

**Trial registration number** ISRCTN30110012.

## INTRODUCTION

To effectively manage chronic obstructive pulmonary disease (COPD), current national[1] and international[2] clinical guidelines recommend the addition of pulmonary rehabilitation (PR) to standard pharmacological therapy. PR is a non-pharmacological intervention known to help manage and improve respiratory symptoms in stable COPD conditions[3] and is also recommended following a hospital admission for acute exacerbation of COPD.[1] Moreover, PR is beneficial to improve exercise capacity,[3 4] breathlessness,[3] quality of life (QoL),[5] anxiety and depression,[6] and reduce healthcare utilisation.[7] However, these benefits may only be sustained for 6–12 months after PR.[4] A number of factors may

affect the ability to maintain exercise capacity and QoL in the long term including: frequency of supervised maintenance exercise; strategies used to improve adherence to maintenance exercise; facilitators and barriers to long-term exercise training; and initial PR programme itself.[8] Repeated exacerbations, progression of the disease and existence of co-morbidities may also have an impact[4] Additionally, from a psychological perspective, patients with COPD report the feeling of abandonment after the graduation from PR as they lose the support from healthcare professionals (HCPs) and peers received during PR.[9]

Many interventions have been introduced to facilitate maintaining the benefits of PR beyond a 6–12 months period. Some of those studies revealed that maintenance interventions following PR is of benefit to COPD clients to maintain PR outcomes in the short term and long term (6–24 months).[10–12] However, the interventions were heterogeneous in terms of duration, setting and level of supervision following initial PR. In addition, patients' motivational and behaviour change strategies play a major role in maintenance but were examined in a limited numbers of the aforementioned maintenance interventions.[10–12]

Behaviour change can be supported using motivational interviewing (MI) techniques. MI is used to help coach people to make personal changes in their behaviour.[13] MI has been shown to outperform traditional 'advice giving' in the treatment of a broad range of behavioural problems/diseases.[14] To our knowledge, none of the previous maintenance interventions have used this technique per se. In COPD, MI coaching delivered on hospital discharge was shown to have a positive influence on readmissions and QoL compared with a control group.[15]

Incorporating an exercise manual with group sessions underpinned by MI techniques has been tested in primary and secondary settings as a home PR alternative compared with usual care[16] and traditional (centre-based) PR.[17] The studies revealed benefits in exercise performance, knowledge, anxiety and fatigue in which some of these outcomes were maintained 6 months after the initial delivery, when compared with usual care[16] and non-inferior to PR.[16] More recently, the programme has been delivered in groups to enhance peer support in COPD.[18]

Thus, for this trial, we propose a supervised maintenance package (Self-management Programme of Activity, Coping and Education FOR COPD, SPACE FOR COPD) with the aim of retaining the expected benefits of PR as well as increasing patients' motivation and to address the feelings of abandonment following traditional PR. We will particularly focus on exacerbation and mood management, as well as promoting regular exercise in our programme. By developing this post PR intervention, we aim to preserve physical and mental well-being, potentially reducing the likelihood of repeat PR attendance and hospital readmissions. In this trial, we hypothesise that patients will have improvements in PR outcomes such as exercise capacity and overall QoL that will be maintained in the intervention group for at least 1 year. This also could be cost effective for the health service.

## Aims

The overarching aim of the study is to assess whether a self-management maintenance programme (SPACE FOR COPD), delivered on PR discharge, preserves the anticipated gains made in exercise tolerance and mental well-being at 12 months, compared with usual care alone.

The primary aim is:
► To compare endurance walking distance at 12 months in those who have completed the SPACE FOR COPD maintenance programme compared with usual care.

The secondary aims are:
► To compare the following variables at 12 months in patients who have completed the SPACE FOR COPD© maintenance programme versus usual care:
  – Maximum walking distance.
  – QoL.
  – Anxiety and depression.
  – Physical activity.
  – Symptoms (breathlessness, fatigue, sputum production, cough, sleep, chest tightness).
  – Patient activation (knowledge, skills and confidence to manage COPD).
  – Forced expiratory volume in 1 s.
► To measure adherence to the SPACE FOR COPD maintenance programme in terms of sessions attended, completion of the programme and manual diary entries.
► To collect data for future health economic analysis in both groups; including hospital admissions, healthcare contacts, personal social service costs and patient-borne expenditure.
► To understand the patient experience of SPACE FOR COPD; to identify facilitators and barriers of the package to maintain health and well-being.
► To work with staff, managers and commissioners involved in the delivery of SPACE FOR COPD as a maintenance option to explore how, if successful, this programme could be rolled out more widely.
► To use data pre-PR, where available, to evaluate the effectiveness of the initial PR programme and its impact on the maintenance rehabilitation programmes outcomes.

## METHODS AND ANALYSIS
### Trial design and setting

The trial will be a prospective, multicentre, randomised controlled, single blind trial of SPACE FOR COPD maintenance programme vs usual care. There will be an integrated economic evaluation and also an embedded qualitative component. The trial will be conducted at two centres, the University Hospitals of Leicester (UHL) National Health Service (NHS) Trust in Leicester, UK, and Harefield Pulmonary Rehabilitation Unit, Harefield, UK. We have used the SPIRIT-2013 checklist and recommendations as a guide for designing this trial.[19]

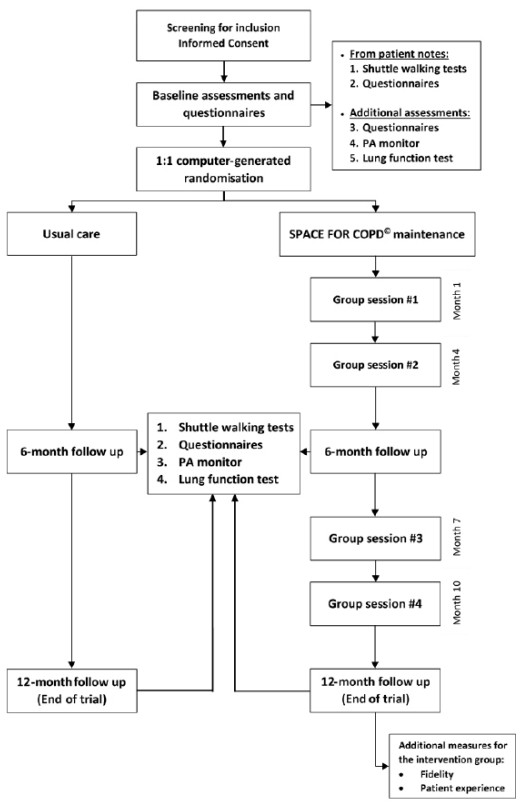

Questionnaires include: Chronic Respiratory Questionnaire-self reported, Chronic Obstructive Pulmonary Disease Assessment Test, EuroQol- five dimensions- five levels, Hospital Anxiety and Depression Scale, and Resource Use Questionnaire.

**Figure 1** Study flow chart. SPACE FOR COPD, Self-management Programme of Activity, Coping and Education for Chronic Obstructive Pulmonary Disease. PA, Physical Activity

The trial will consist of three visits (baseline, 6 and 12 months). Each visit will last for up to 2 hours where all outcomes will be collected. In addition, a 2-hour small group session will be conducted at month 1 after baseline visit then at months 4, 7 and 10 in the intervention group only (see figure 1). The trial recruitment will take place over 1 year, with a 1-year follow-up.

## Participants

### Eligibility criteria for patients

Participants will be included if they meet the following criteria:

► Completed PR within the last 4 weeks (PR must meet standards defined by the British Thoracic Society (BTS).[20]
► Clinical diagnosis of COPD.
► Able to read and write English to the age of an 8 year old.
► Eighteen years and older.
   Participants will be excluded if they:
► Have a significant disability which limits the daily physical activity.
► Unable to read and write English due to the nature of the manual's language.

### Eligibility criteria for HCPs

HCPs involved in delivering the intervention and will be invited to participate in focus groups. The inclusion criterion is to deliver one or more sessions. The aim is to explore their experience, the barriers and facilitators in delivering the group sessions. The HCPs are respiratory physiotherapists, respiratory nurses or health psychologists with experience in delivering PR. All involved HCPs have received MI training delivered by a health psychologist and introduction on how to deliver the group sessions using a dedicated trial manual.

## Procedure

### Usual care (control group)

The control group will receive usual care: written maintenance advice as recommended by the BTS quality standards.[21] Leaflets will be given to signpost the individual to maintenance options for exercise following PR. A referral to community exercise schemes will be arranged for the control and intervention groups, if available or desired, as this represents best usual care across the UK.

### Space for COPD maintenance programme (intervention group)

Participants in the intervention group will receive the SPACE FOR COPD home-based maintenance programme manual and be invited to attend SPACE FOR COPD group-based maintenance sessions detailed below.

### Space for COPD home-based manual

Precise details of the rationale, development and efficacy of the work underpinning the SPACE FOR COPD manual are available elsewhere.[22] The manual is divided into four stages providing an exercise programme and covering educational topics. It includes self-management skills, disease specific information and tasks. The content of the manual has been approved by the Plain English Campaign and received the Crystal Mark for Clarity of British English. The content of the manual is outlined in table 1. Table 2 demonstrates the mapping of self-management components of the manual to the PRISMS taxonomy framework. There is a COPD action plan insert which is consistent with the recruiting centres' usual care.

For the purpose of this trial, a new single page insert has been developed to facilitate longer term goal setting by following the Specific, Measurable, Achievable, Relevant and Time bound method for goal settings. HCPs will advise participants on how to use and follow the manual independently at home. Participants will also be asked to complete section of the manual at home (eg, exercise diaries) and will have the opportunity to call HCPs with questions via telephone.

### Group sessions

The group sessions are held in small groups (5–10 participants per group) that will accompany the SPACE FOR COPD maintenance programme manual with the aim of discussing progress, addressing barriers, increasing motivation to maintenance, to enhance health lifestyle behaviours and to encourage social support. The content

| Table 1 | Manual components |
|---|---|
| **Stage 1** | **Stage 4** |
| What's happened to your lungs? | Staying fit and your hobbies |
| Exercise: How to get fitter | Your relationships |
| Setting your goals | Dealing with setbacks |
| Managing your stress - part 1 | Sex and your lungs |
| Your emotions | Breathe easy |
| Controlling your breathing | Conclusion |
| Information about your medication | |
| **Stage 2** | Frequently Asked Questions (FAQs) and appendix |
| How to stay fit | Frequently asked questions |
| Managing days when you feel unwell | Setting your walking speed |
| Saving your energy | Help for carers |
| Diet and feeling unwell | Advice about oxygen |
| Advice for clearing your chest | Smoking: advice on giving up |
| **Stage 3** | Cleaning your breathing devices |
| How to get stronger | Spare walking diaries |
| Managing your stress—part 2 | Spare strength training diaries |
| Healthy eating | |
| Travelling and your lung disease | |

of these sessions is shown in figure 2. HCPs facilitating the groups will use MI techniques to identify barriers and facilitators to maintaining exercise, expressing empathy, supporting self-efficacy and 'rolling with resistance'. Group sessions will be facilitated by two HCPs (eg, Physiotherapists, respiratory specialist nurses, health psychologists). There will be four sessions every 3 months over the period of 1 year.

We anticipate that some participants may not be able to attend all the group sessions or may be unable or unwilling to attend the research centre. In this instance, we will accommodate the participant in another session (with permission of that group) or via videoconferencing. If this isn't possible a telephone contact will be conducted instead.

### Staff training and intervention fidelity
In total, eight HCPs have been trained to deliver the SPACE FOR COPD maintenance programme by a health psychologist and a specialist nurse. All attended a 1-day training course (supplemented with written material and DVD tutorials to take away) to learn the structure of the structure of the group sessions, the SPACE for COPD manual and MI skills.

The agenda covered: MI skills, introduction to the facilitator manual and review of SPACE FOR COPD manual, practical workshop, group facilitation, learning outcomes revisited and trial management.[6] of the HCPs had previously received additional MI training and all had experience in delivering PR.

A teleconference will be held between all the facilitators across the two sites prior to and following the first intervention session to problem solve and share experiences. The teleconference will then be continued on a regular basis to maintain the flow of the maintenance procedure and to provide peer support.

Quality assurance will be undertaken to assess delivery of intervention content and educational style. Intervention fidelity checklists for the intervention facilitators have been adapted from a previous study.[18] One session per self-management group will be peer-observed and the checklist will be completed. Each site will observe the opposite site's group. Facilitators will complete fidelity checklists at the end of each session. These will assess: content covered, managing groups and adhering to MI principles.

For the trial, external auditing may take place at the discretion of the sponsor (UHL NHS trust).

### Study outcomes
The assessments described below will be conducted at baseline, 6-month and 12-month visits and completed by a blinded assessor, unaware of the group allocation. Due to the impact of COVID-19 restrictions, assessment visits and consent may be completed over the phone for all data other than physical tests where this is not possible (ie, spirometry/walking tests). Consent form for patients shown in online supplemental file 1.

### Primary outcome
#### Endurance capacity
The primary endpoint at 12 months is endurance capacity measured by the Endurance Shuttle Walk Test (ESWT), expressed in seconds. The ESWT is a walking field test that has been found to be safe, valid and reproducible among patients with COPD.[23] The minimal clinically important difference (MCID) for this population is estimated to be between 174 and 279 s.[24]

### Secondary outcomes
#### Exercise capacity
Incremental Shuttle Walking Test (ISWT) will be used to measure the maximal exercise capacity following the test protocol published by Singh *et al*.[25] The MCID in walking distance is between 35.0 and 36.1 m.[26]

#### Chronic Respiratory Questionnaire-self reported
Chronic Respiratory Questionnaire-self-reported (CRQ-SR) is a valid and reliable self-administered tool to assess health-related QoL (HRQoL) in people with chronic lung diseases.[27] It assesses four domains with MCID defined as a mean change of 0.5 for each domain. The domains are dyspnoea (five questions), fatigue (four questions),

**Table 2** Practical Reviews in Self-Management Support (PRISMS) taxonomy components and examples in the manual

| PRISMS taxonomy component | Elaboration | Examples from the SPACE FOR COPD facilitator manual |
|---|---|---|
| Information about condition and/ or its management | Specific information about COPD is provided within the Stage 1 of the manual. | Participants are provided with information throughout the programme (every session) |
| Information about available resources | | Participants are provided with information throughout the programme (every session) |
| Provision of/agreement on specific clinical action plans and/or rescue medication | COPD Action Plan is included as an insert in the manual | Actions plans in session 2 |
| Practical support with adherence (medication or behavioural) | Walking and strength training diaries are provided for participants and discussed during sessions | Walking and strength training diaries are provided for participants and discussed during solution focused goal feedback at the beginning of sessions 2, 3 and 4 |
| Provision of easy access to advice or support when needed | Participants are able to call programme facilitators between sessions with questions | Participants are provided with contact details for programme facilitators who the can call if needed (throughout programme). |
| Training/rehearsal for practical self-management activities | Including:<br>► Managing exacerbations<br>► Saving your energy | Session two covers managing exacerbations<br>Session three covers saving your energy |
| Training/rehearsal for psychological strategies | Including:<br>► Goal setting<br>► Solution focused goal feedback<br>► Problem solving<br>► Self-reward and social reward<br>► Managing stress and emotions | Goal setting activity (including action planning) and solution focused goal feedback (once every session).<br>Problem solving (throughout programme)<br>Managing stress and emotions in session 3. |
| Social support | Including:<br>► Practical support<br>► Emotional support<br>► Peer support<br>► Group socialising is encouraged | Participants are encouraged to share experiences, advice, ideas and support each other (throughout the programme). |
| Lifestyle advice and support | Including:<br>► Introduction to the walking programme<br>► Strength training<br>► Hobbies<br>► Maintaining exercise | Introduction to walking programme in session 1<br>Hobbies (session 3)<br>Maintaining exercise (session 4) |

SPACE FOR COPD, Self-management Programme of Activity, Coping and Education for Chronic Obstructive Pulmonary Disease.

emotional function (seven questions), and mastery (four questions) and are scored on a 7-point scale per question.

### Chronic obstructive pulmonary disease assessment test

Chronic assessment test (CAT) is a tool developed to assess and monitor COPD symptoms and changes in patient's health status.[28] It is responsive in both short-term and long-term assessments and is used widely in PR programmes for patients with COPD.[4] It consists of eight simple questions which cover 'cough, phlegm, chest tightness, breathlessness going up hills/stairs, activity limitation at home, confidence leaving home, sleep and energy' on a 5-Likert scale. Scores ranges from 0 to 40 indicating none to very severe impact on health status.

### EuroQol-five Dimensions-five Levels

The EuroQol-5 Dimensions-5 Levels (EQ5D5L) is a self-administered non-disease specific tool used to measure HRQoL and is developed by the EuroQol group.[29] This measures health on five dimensions (mobility, self-care, usual activities, pain or discomfort, and anxiety or depression) and a tariff is available for deriving a single utility score based on time trade-off utility scores. It also includes a visual analogue scale on which individuals rank their health today from 0 (worst health) to 100 (best health).[30]

### Patient activation measure

The NHS long-term plan[31] supports the engagement of patients to self-management, which is a vital key to our trial, where the 13-item patient activation measure

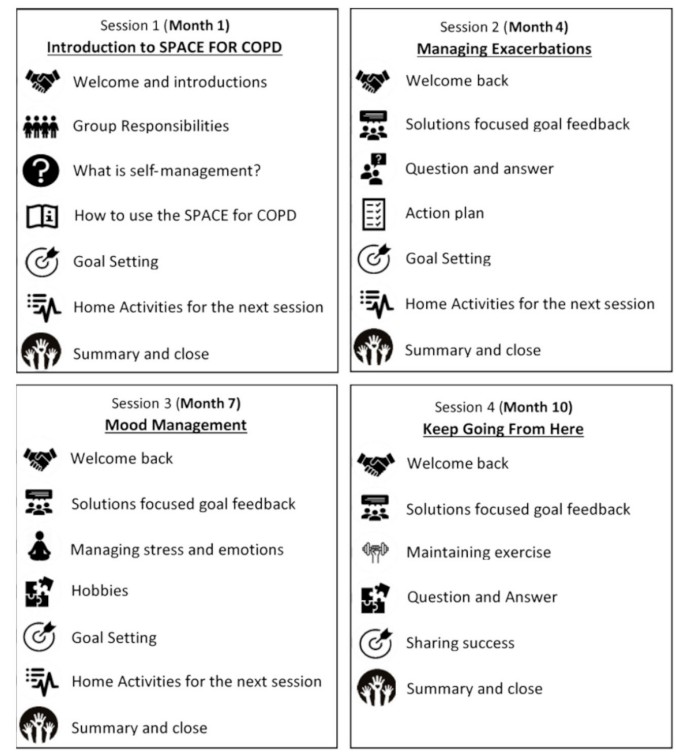

**Figure 2** Group sessions contents. SPACE FOR, COPD, Self-management Programme of Activity, Coping and Education for Chronic Obstructive Pulmonary Disease.

(PAM-13) tool can be used as recommended by NHS.[32] PAM-13 tool assesses participants' knowledge, skill and confidence for managing their own health and health-care.[33] It is valid and reliable tool that defines four levels of activations ranges from low level to high level of activation.

## Anxiety and depression
The Hospital Anxiety and Depression Scale (HADS) will be used to measure anxiety and depression.[34] It is a responsive tool and it has been tested on patients with COPD in several study settings.[35] It has 14 questions to measure anxiety and depression (seven questions for each scale). Minimum and maximum scores for each scale are 0–21, respectively. The MCID is a mean change of 1.5 points for each scale.[36]

## Physical activity monitoring
The participants will be asked to wear a physical activity monitor after each visit (wGT3X- ActiGraph, Pensacola, Florida, USA), for seven consecutive days, except during sleeping and water-based activities. The monitor will be activated automatically for that period and doesn't require the participant to turn it on or off. It is a small device attached to an elastic belt to be worn on the waist and placed at the mid right hip. It will enable us to retrieve daily activity bouts, intensity of activities and step counts. Activity monitors can be posted out if assessment is performed over the phone.

## Pulmonary function testing
Pulmonary function will be assessed by hand-held spirometry called Vitalograph Pneumotrac 6800 (Vitalograph, Buckingham,UK) to confirm and assess the severity of COPD. The test will be performed if there was no previous test done within a 3 month period. The spirometry test will be done following standard guidelines.[37] The device will be calibrated on daily basis and participants will be assessed for contraindications prior to performing the test.

## Patient adherence
The exercise diary entries embedded within the SPACE FOR COPD manual will be used to measure adherence to the home-based maintenance programme part. Also, group session attendance will be recorded and calculated as total attendance rate. A 75% cut-off will be considered as a programme completion at the end of the trial. To reduce lost to follow-up, we will reimburse participants' travel expenses. Where patients drop-out of the SPACE FOR COPD maintenance programme, we will still collect 6-month and 12-month assessment data if consent is still in place.

Adherence to the programme is critical. To improve adherence we will be as flexible as possible with sessions and closely monitor home exercise diaries. If a participant is unable to attend a session we will offer attendance of the same session within another group (with the verbal consent of that group) or via teleconference as the preferred option or a one to one contact. We will send out text/email/postal reminders for group sessions and appointments. We expect that some patients may readmit to hospital and attend PR again. This will be documented but we would not deny further postexacerbation PR in these patients, in accordance with UK guidelines.[20]

## Resource use questionnaires
Structured resource use questionnaires have been designed for the purpose of this trial and piloted prior to use. These will collect information on the number of hospital inpatient and outpatient episodes; community health and social care costs; and changes to employment status. Patient-borne costs associated with hospital visits (eg, travel, dependant care, time off work) and out-of-pocket expenditure on products or equipment (eg, walking shoes) will also be recorded. Questionnaires will be administered to coincide with visit times that is, baseline, 6 months and 12 months.

## Interviews with patients
Semistructured, face-to-face interviews will be conducted with patients in the intervention group: both completers and non-completers. This will be optional and there will be a separate box on the patients' consent form for participants to opt in (online supplemental appendix 1). The aim of the interview is to give a more complete picture of the perspective of the subjects of the study[38]; we will be adopting an interpretativist approach.[39] At the end

of the 12-month period (or before if a patient decides not to continue the group sessions), we will conduct an interview to explore their experiences of using the new maintenance manual at home, the group sessions and the research process.

Participants will be recruited using a purposeful sampling technique: maximum variation sampling strategy. These interviews (n=10–20 or until data saturation) will be organised and completed by a qualitative researcher who has not been involved in delivering the intervention. Indicative interview schedules will be used and so may be refined following the first 4–5 interviews. The purpose of this qualitative work is to explore perceptions of SPACE FOR COPD as a maintenance strategy.

We will explore (Intervention) patients' experiences of SPACE FOR COPD, identifying barriers and facilitators to the maintenance of well-being facilitated by the programme and to examine whether the sense of abandonment is lessened. The interview schedule will be flexible enough to allow completely unanticipated issues to arise. The interviewer will have adequate experience and expertise to deal with difficult situations (eg, distress).

The patient interviews will take place at the point of trial completion (at 12 months or earlier if there is attrition). We are interested in the views of patients who do not complete the programme also. Interviews will take place either in hospital or in the patient's home or via phone or video call for those unable to attend the research centre.

### Focus group for HCPs
We will also conduct focus groups with staff involved in delivering the intervention, to understand the relative burden and benefits of this new intervention to their services. Two focus groups with 5–10 participants per group will be organised and conducted by a qualitative researcher who has not been involved in delivering the intervention.

Staff will attend a semistructured focus group at their workplace and the findings of these will inform the training needs of those providing the intervention. They will also help to ensure that the intervention is moulded in a way that is compatible with existing models of service provision and working practices, which may help to maximise the likelihood of uptake locally and nationally.

### Pre-PR data
Baseline data from the initial PR will be collected, where available, to evaluate the effectiveness of the PR and its impact on the SPACE FOR COPD maintenance programme. Data will include: ESWT, ISWT, CRQ-SR, CAT, HADS as these measures are routinely collected as part of PR.

### Sample size
The power calculation (for a continuous outcome superiority trial) is based on the primary outcome measure. We require 116 patients for an 80% chance of finding a 184s (SD: 282s) difference in the ESWT time between groups at 12 months (at the 5% level). The 184s was the between group difference reported in the original primary care SPACE FOR COPD study[14] and is similar to the suggested MCID range for this outcome (174–279s).[24] This number includes expected attrition of 20%.

### Recruitment and invitation of participants
Participants will be approached and assessed for eligibility on completion of PR by members of the clinical PR services. Interested participants will receive a participant information sheet and given at least 48 hours to consider their participation. After consultation with our patient and public involvement (PPI) group, we propose not to set a threshold for improvement in PR outcomes for patients to be considered for the trial. Those interested will receive an invitation letter to attend a consent research visit.

### Randomisation and blinding
Eligible participants will be consented and randomised using a concealed allocation web-based programme (https://www.sealedenvelope.com/). Randomisation will be 1:1 to either usual care (control) or the SPACE FOR COPD maintenance programme (intervention). Randomisation codes will be blocked per site, providing even numbers of control and intervention patients at each site and a sufficient number of participants for the intervention group sessions to run. Due to the nature of the intervention, the participants and intervention facilitators will not be blinded to group allocation. However, the assessors will be blinded to group allocation. Other blinded assessors are in place in case of blinding exposure by first blinded assessor. Any unintentional unblinding will be documented.

### Data collection and management
The REDCAP (Research Electronic Data Capture) database housed by University of Leicester will be used to create our paper case report forms (CRFs) and transfer data to the electronic version will be performed periodically. This password-protected database has the facility to set limits, flag discrepancies and provides an audit trail of when data has been entered/amended and by whom. A random check of 10% of the data will be performed by a member of the research team not involved in data entry. Once data are ready for analysis, it can be easily exported to SPSS statistical analysis software. Paper CRFs will not contain patient identifiable data and will be stored in locked filling cabinets in key coded offices. Data will be collected and stored in accordance with the 2018 UK General Data Protection Regulation.

### Data analysis
#### Quantitative analysis
All data will be analysed on an intention-to-treat basis using SPSS software (V.26.0, IBM). We will conduct a separate per-protocol analysis for those who complete the intervention. Continuous variables will be presented as mean (SD), categorical variables as frequencies (percentages).

All data will be assessed for outliers, normality and analysed using appropriate parametric and non-parametric statistics. To test the primary outcome, the difference between the two groups for the ESWT time will be analysed at 12 months using an independent t-test (or non-parametric equivalent).

Secondary outcome measures will be analysed in the same way. We will also look at changes in all outcomes in individual participants over time (ie, pre-PR, post-PR, 6 months and 12 months) using a repeated measures analysis of variance (or non-parametric alternative). Significance will be set at $p < 0.05$ and differences between groups will be expressed as mean (95% CIs). Analysis will take into account the baseline values and adjust for potential confounders (eg, age, disease severity). Missing data will be corrected for by multiple imputation. Interim data may be made available for abstracts but unmasking will not occur.

### Economic analysis

An economic analysis will be performed from a societal perspective (including patientborne costs) and from a health and social care perspective. Patient use of NHS services will be priced using Personal Social Services Research Unit (PSSRU[40]) costings or NHS reference costs[41] and include appropriate on-costs. Impact on outcomes (health utility) will be assessed using the EQ5D questionnaire. A cost–utility analysis will compare incremental costs and marginal benefits, and include 1000 bootstrapped replication.[42]

The cost per quality-adjusted life-year (QALY) gained, relative to standard care, will be calculated. Intervention costs will be apportioned fully to the 12-month period. A cost-effectiveness acceptability curve will be generated to indicate the probability that the intervention is cost-effective at various cut-off thresholds for the societal valuation of a QALY.[43] Sensitivity analyses will be undertaken.

### Qualitative analysis

All interviews/focus groups will be recorded and transcribed verbatim with participants' written consent. Data will be analysed according to the framework analysis approach.[44] This approach allows for data collection to start with preset objectives while ensuring that analysis reflects the experiences of participants. Framework analysis follows five stages: familiarisation with data, identification of thematic framework, indexing of data to thematic framework, charting of data to framework and interpretation of the framework.

### Harms

Patients will be asked to complete several simple physical tests at the trial visits. They may feel breathless/tired at the end of the walking tests but this is normal and usually settles within a few minutes. It would be very unusual for symptoms to persist. However, in this unlikely situation, medical staff are on hand at the hospital. We follow a standard operating procedure for walking tests to ensure

safety. If any results from the tests undertaken as part of the trial are clinically significant, we will inform both the participant and their general practitioner (with patient consent).

We will abide by the safeguarding policy of the University Hospitals of Leicester NHS Trust and Royal Brompton & Harefield NHS Trust to ensure that all staff working on the trial have undertaken safeguarding training.

### PPI statements

For this trial, we have two PPI members on the project steering committee, who will be involved in all stages of the research cycle. We have one member from each recruiting site to ensure that local issues may be addressed. We know from previous projects how valuable our PPI colleagues can be in terms of helping to shape the research and troubleshooting any issues. For instance, our recent SPACE FOR COPD pragmatic study (delivering the SPACE FOR COPD programme to community groups) benefited from PPI feedback when designing the format and scheduling of the patient group sessions to ensure that the travel arrangements and times were not onerous for patients.

For this project, PPI members have helped us to adapt the protocol in in terms of agreeing the participant inclusion criteria and how the SPACE FOR COPD maintenance programme should be delivered. The relationship between researchers and PPI members will be collaborative; the reason for this is to ensure that there is an ongoing partnership between the research team and the service users, where decisions about the research are shared. To help support members, we will coproduce all documentation to ensure they are written in lay language. We anticipate that the PPI members will be involved in developing and reviewing patient documents.

We would involve different stakeholders such as other health professionals, commissioners and charities (eg, the British Lung Foundation Breathe Easy groups) in our discussions. We provide training to our PPI members and have budgeted appropriately to support this level of PPI taken from the INVOLVE guidelines.[45] PPI colleagues will be involved in the dissemination of the work at a local level.

## ETHICS AND DISSEMINATION

The trial was granted ethical approval from Health Research Authority and Health and Care Research Wales (HCRW) with Research Ethics Committee number 19/EM/0267 on 10 October 2019. The first participant was recruited on 18 October 2019. The study is sponsored by University Hospitals of Leicester NHS Trust and funded by National Institute for Health Research with grant number PB-PG-0317-20032. Amendments to protocol will go through HCRW and sponsor before enforcement.

In order to have a well-structured dissemination plan, we have adopted the Scientist Knowledge Translation Plan.[46] It guides researchers through the stages of developing a dissemination strategy including clarifying 'key

messages' identifying ways to transmit those messages and designing an evaluation plan. See here for details: http://ktdrr.org/training/webcasts/webcast5/webcast_ktplan_101013r.pdf

Audiences for this trial are commissioning organisations [such as Clinical commisioning groups (CCGs) and NHS England), HCPs providing rehabilitation, patients and the public, external statutory organisations (such as the Department of Health, National Institute for Health and Care Excellence, Applied Research Collaborations (ARCs) and Academia. They will be involved throughout the research process to better shape the outcomes and have it easily translated to the community at the end of the trial. The delivery of the trial outcomes will be communicated via face to face. Moreover, it will include, but not limited to, development of links with key organisations, use of electronic media and publications including full and plain English summary reports.

**Author affiliations**
¹Respiratory Therapy Department, Jazan University, Jazan, Saudi Arabia
²Respiratory Sciences, University of Leicester, Leicester, UK
³Centre of Exercise and Rehabilitation Science, Leicester Biomedical Research Centre- Respiratory, Glenfield Hospital Respiratory Medicine Department, Leicester, UK
⁴Department of Respiratory Medicine, Royal Brompton and Harefield NHS Foundation Trust, London, UK
⁵Faculty of Health and Life Sciences, Coventry University, Coventry, UK
⁶Faculty of Medicine, Imperial College London, London, UK

**Contributors** KAA, CG, CMN, NG, AS, WM, SJS and LH-W all members of the study steering group involved in design and conduct of the study, collecting data and delivering the study procedures.

**Funding** This work was funded by the National Institute for Health Research (NIHR) Research for Patient Benefit Programme, grant number (PB-PG-0317-20032) and supported by the NIHR Leicester Biomedical Research Centre (BRC)- Respiratory.

**Competing interests** None declared.

**Patient and public involvement** Patients and/or the public were involved in the design, or conduct, or reporting, or dissemination plans of this research. Refer to the Methods section for further details.

**Patient consent for publication** Not applicable.

**Provenance and peer review** Not commissioned; externally peer reviewed.

**ORCID iDs**
Khaled A Alqahtani http://orcid.org/0000-0001-5235-8467
Claire M Nolan http://orcid.org/0000-0001-9067-599X
Ala Szczepura http://orcid.org/0000-0001-6244-9872
Linzy Houchen-Wolloff http://orcid.org/0000-0003-4940-8835

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
