## [Reviewer comments · BMJ Open]

ARTICLE DETAILS

TITLE (PROVISIONAL)	SPACE FOR COPD© delivered as a maintenance programme on Pulmonary Rehabilitation discharge: protocol of a randomised controlled trial evaluating the long-term effects on exercise tolerance and mental wellbeing.
AUTHORS	Alqahtani, Khaled A.; Gerlis, Charlotte; Nolan, Claire; Gardiner, Nikki; Szczepura, Ala; Man, William; Singh, Sally; Houchen-Wolloff, Linzy

VERSION 1 – REVIEW

REVIEWER	Dominic Dellweg Fachkrankenhaus Kloster Grafschaft GmbH, Abteilung Pneumologie, Pulmonology 1
REVIEW RETURNED	12-Sep-2021

GENERAL COMMENTS	Linzy Houchen-Wolloff and coworkers describe their protocol for a post PR home based intervention program for COPD patients (SPACE FOR COPD © (Self-management Programme of Activity, Coping, and Education) is a home-based self-management programme). The protocol is well written and of good structure. My remarks are as follows. 1. Time plan: Recruiting and follow up period are one year. A specific date for the beginning of the trial is not provided.2. Study outcomes (page 10 line 45): ' Study outcomes: The assessments described below will be conducted at baseline, 6- and 12-month visits and completed by a blinded assessor, unaware of the group allocation. Due to the impact of COVID-19 restrictions, assessment visits and consent may be completed over the phone.' Assessments however includes PFT's and endurance tests which can hardly be done over the phone. Please clarify.3. Primary outcome (page 10 line 53 ff): The reference which is provided for the MCID for the Endurance Shuttle Walk Test (ESWT) is '24 Evans RA, Singh SJ. Minimum important difference of the incremental shuttle walk test distance in patients with COPD. Thorax. 2019;74(10):994-5.' This study however evaluated the incremental shuttle walking test (ISWT)4. Disease severity (PFT, endurance test results etc.) has to match the cohort by Zatloukal Jet al. (Chronic Respiratory Disease. 2019;16:1479973119853828), otherwise the applied MCID might not apply.5. Randomisation (page 15 line 26): Randomisation does not take any parameters such as basic demographic data, functional data, the use of oxygen or smoking status into account. The authors are encouraged to conduct a matched randomization.6. Data collection and management (page 15 line 46): Besides a 10 % random data check an additional check should be done to find
--

	outliers. 7. Data analysis (page 16 line 8 ff.): Comparison of ESWT time at 12 month is not helpful at all since the baseline might have been different. If the authors want to avoid multiple testing (which I would recommend but which would require another power calculation) they might compare the 12 month difference in ESWT. Since assessments are done at 0, 6 and 12 month I would recommend ANOVA with repeated measurements.
--	--

REVIEWER	Lissa Spencer Sydney Local Health District, Physiotherapy
REVIEW RETURNED	02-Nov-2021

GENERAL COMMENTS	Thank you for your interest in Maintenance programs following pulmonary rehabilitation and your hard work in preparing this protocol. Pulmonary rehabilitation has been shown to be part of the ideal management for people with COPD and other respiratory conditions. Maintaining the benefits long term is difficult to achieve and this protocol may shed some light on a difficult area of management. You have written a very thorough protocol and have considered absolutely everything. I wonder if you may have benefited from a couple of references:  • Maintaining the benefits following pulmonary rehabilitation: Achievable or not? Spencer LM, McKeough ZJ. • Maintaining benefits following pulmonary rehabilitation: a randomised controlled trial. Spencer LM, Alison JA, McKeough ZJ. Thank you again for your hard work and good luck in the future
---

VERSION 1 – AUTHOR RESPONSE

Reviewer: 1

Dr. Dominic Dellweg, Fachkrankenhaus Kloster Grafschaft GmbH, Abteilung Pneumologie

Comments to the Author:

Linzy Houchen-Wolloff and coworkers describe their protocol for a post PR home based intervention program for COPD patients (SPACE FOR COPD © (Self-management Programme of Activity, Coping, and Education) is a home-based self-management programme).

The protocol is well written and of good structure.

My remarks are as follows.

1. Time plan: Recruiting and follow up period are one year. A specific date for the beginning of the trial is not provided. The start date (date of ethical approval) is already provided at the end of the abstract and in the ethics and dissemination section of the paper.

2. Study outcomes (page 10 line 45):' Study outcomes: The assessments described below will be conducted at baseline, 6- and 12-month visits and completed by a blinded assessor, unaware of the group allocation. Due to the impact of COVID-19 restrictions, assessment visits and consent may be completed over the phone.' Assessments however includes PFT's and endurance tests which can hardly be done over the phone. Please clarify. We have added the following to clarify: *Due to the impact of COVID-19 restrictions, assessment visits and consent may be completed over the phone for all data other than physical tests where this is not possible (i.e. spirometry/ walking tests).*

3. Primary outcome (page 10 line 53 ff): The reference which is provided for the MCID for the Endurance Shuttle Walk Test (ESWT) is '24 Evans RA, Singh SJ. Minimum important difference of the incremental shuttle walk test distance in patients with COPD. *Thorax*. 2019;74(10):994-5.' This study however evaluated the incremental shuttle walking test (ISWT) Apologies but the reference list was not in the correct order. This has been updates and the correct reference is 24 (Zatloukal).

4. Disease severity (PFT, endurance test results etc.) has to match the cohort by Zatloukal Jet al. (*Chronic Respiratory Disease*. 2019;16:1479973119853828), otherwise the applied MCID might not apply. This population is the same as our cohort (i.e. from the same recruiting centre). This is the only available MCID for the ESWT from a rehab population.

5. Randomisation (page 15 line 26): Randomisation does not take any parameters such as basic demographic data, functional data, the use of oxygen or smoking status into account. The authors are encouraged to conduct a matched randomization. We are using a simple randomisation system, if the groups are not balanced this will be accounted for in the analysis. This is discussed already ion the quantitative data analysis section: *Analysis will take into account the baseline values and adjust for potential confounders (e.g. age, disease severity)*.

6. Data collection and management (page 15 line 46): Besides a 10 % random data check an additional check should be done to find outliers. As well as the 10% random data check (at both sites), the database we are using has the facility to set limits and flag discrepancies. We have added to the quantitative data analysis section: *All data will be assessed for outliers, normality and analysed using appropriate parametric and non-parametric statistics*.

7. Data analysis (page 16 line 8 ff.): Comparison of ESWT time at 12 month is not helpful at all since the baseline might have been different. If the authors want to avoid multiple testing (which I would recommend but which would require another power calculation) they might compare the 12 month difference in ESWT. Since assessments are done at 0, 6 and 12 month I would recommend ANOVA with repeated measurements. The primary outcome is at 12months because the aim of the study is to look at the long term sustainability of the ESWT and the intervention is delivery over 12months. We have added in repeated measures ANOVA to our quantitative data analysis section: *We will also look at all outcomes in individual participants over time (i.e. pre-PR, post-PR, 6-months and 12-months) using a repeated measures ANOVA (or non-parametric alternative)*.

Reviewer: 2

Dr. Lissa Spencer, Sydney Local Health District

Comments to the Author:

Thank you for your interest in Maintenance programs following pulmonary rehabilitation and your hard work in preparing this protocol. Pulmonary rehabilitation has been shown to be part of the ideal management for people with COPD and other respiratory conditions. Maintaining the benefits long term is difficult to achieve and this protocol may shed some light on a difficult area of management. You have written a very thorough protocol and have considered absolutely everything. I wonder if you may have benefited from a couple of references:

- Maintaining the benefits following pulmonary rehabilitation: Achievable or not?

Spencer LM, McKeough ZJ. I have added this reference and the following text, thank you: *A number of factors may affect the ability to maintain exercise capacity and QoL in the long term including: frequency of supervised maintenance exercise; strategies used to improve adherence to maintenance exercise; facilitators and barriers to long-term exercise training; and initial PR programme itself (8)*.

- Maintaining benefits following pulmonary rehabilitation: a randomised controlled

trial. Spencer LM, Alison JA, McKeough ZJ. We have not discussed individual trials in the background, in favour of including the 3 systemic reviews on this topic instead which encompass all trials.

Thank you again for your hard work and good luck in the future